# Development and reliability testing of a qualitative observational rating system for individuals with brachial plexus injury performing functional capacity evaluation tests

**Tallie M. J. van der Laan**◉*, **Sietke G. Postema**◉, **Corry K. van der Sluis**◉,
**Michiel F. Reneman**◉

University of Groningen, University Medical Center Groningen, Department of Rehabilitation Medicine, Groningen, The Netherlands

◉ These authors contributed equally to this work.
* t.m.j.van.der.laan@umcg.nl

## Abstract

### Background

Individuals with brachial plexus injury may be more prone to develop musculoskeletal complaints, due to compensatory strategies. Quantifying compensatory strategies of these individuals may help to minimize the use of dysfunctional compensatory strategies and prevent musculoskeletal complaints.

### Purpose

To develop and to explore the feasibility of an observational rating system for rating postures and movements of the shoulders and trunk in individuals with brachial plexus injury during the performance of the functional capacity evaluation one-handed-individuals (FCE-OH) and to explore the interrater and intrarater reliability of this rating system.

### Study design

Psychometric study including development and reliability testing.

### Methods

Individuals with brachial plexus injury (n = 15) and able bodied controls (n = 21) were videotaped during the performance of five FCE-OH tests. Abnormal shoulder and trunk movements and postures were identified. The rating system was developed pilot tested and adjusted. The interrater and intrarater reliability of the final draft were determined. Sixteen raters performed two rating sessions, two weeks apart and rated 40 video fragments. Absolute percentages of agreement, kappa (κ) and 95% confidence intervals (CI) were calculated. Feasibility was explored using a questionnaire.

**Data availability statement:** The data underlying the results presented in the study are available from https://doi.org/10.17605/OSF.IO/8JBPY. Videos cannot be shared publicly because of privacy reasons, in accordance with the UMCG research code. Videos are available upon request from research support CvR (contact via researchsupport@cvr.umcg.nl), for researchers who meet the criteria for access to confidential data.

**Funding:** The author(s) received no specific funding for this work.

**Competing interests:** The authors have declared that no competing interests exist.

## Results

The interrater reliability of the rating system was: first session κ = 0.48, 95%CI = 0.36–0.60; second session κ = 0.59, 95%CI = 0.45–0.72. The intrarater reliability was κ = 0.64, 95%CI = 0.50–0.70. Half of the raters agreed that the system was easy to use in clinical practice.

## Conclusions

A rating system for measuring postures and shoulder and trunk movements of individuals with brachial plexus injury was developed. The reliability appeared to be sufficient when the system was applied by the same rater. The interrater reliability was moderate for both rating sessions. The reliability of both overhead lifting tests of the FCE-OH was low. Variation in movement patterns due to differences in the remaining function of the affected arm may have caused difficulties in rating.

## Introduction

Individuals with brachial plexus injury (BPI) have loss of function of their affected arm caused by damage to the brachial plexus. The brachial plexus is a complex network of nerves, origination form the spinal cord in the neck, through the axilla to the limb, controlling movement and sensation of the entire arm and hand. It exists of three trunks, the upper trunk which consists of roots C5 and C6, the middle trunk (root C7), and the lower trunk which is formed by roots C8 and T1. Injury to the brachial plexus leads to diminished motor and sensory function and thereby causes complete or partial loss of arm and/or hand function depending on the affected roots and the severity of the BPI.

Individuals with (BPI) need to compensate for the loss of function of their affected arm. Little is known about the compensatory strategies used by individuals with BPI to perform daily activities and work. Compensation may involve increasing the load on the unaffected arm and using compensatory trunk and/or shoulder movements [1–4]. Compensatory movements may make individuals with BPI more prone to develop musculoskeletal complaints (MSCs), because they may be more exposed to biomechanical risk factors for MSCs such as static muscle contractions, forceful and repetitive movements and working in awkward positions [5]. Approximately half of the individuals with BPI experience MSCs [6], compared with one-third of the general population [7]. Most often the neck and unaffected shoulder are affected [6]. Pain, neuropathic or due to MSCs, affects the quality of life, and job choices in individuals with BPI [8–10].

After acquiring BPI only 45% to 66% of the individuals are able to resume their previous work, and of these individuals, 31% need to adjust or to change jobs after injury [11–14]. A functional capacity evaluation (FCE), is a standardized assessment containing work-related tasks and is used to evaluate an individual's capacity in order to make recommendations for participation in work, while considering the individual's body functions and structures, environmental factors, personal factors, and



health status [15]. By matching an individual's functional capacity to physical work demands musculoskeletal complaints may be prevented. The functional capacity evaluation one-handed (FCE-OH) is a short-form functional capacity evaluation adapted for the use in one-handed individuals and upper limb prosthesis wearers [16]. The results of FCE test are expressed in a quantitative manner, for example how much weight an individual can lift, or the time needed for 30 reaching movements. However, how an individual gets to a certain test result is also important. For example, an individual may use a lot of compensatory movements and thereby be more at risk of developing MSCs. A qualitative score, that evaluates movement patterns of individuals with BPI during the performance of the FCE-OH tests may improve return-to-work recommendations based on the FCE-OH test results. Also, it may help professionals to develop treatment plans to minimize the use of dysfunctional compensatory movements during the rehabilitation treatment. This is important because it is difficult to replace a learned dysfunctional movement with a more optimal movement, and the use of dysfunctional compensatory movements may lead to more disability [17]. Furthermore, individuals with BPI get insight into their movement patterns and movement habits using this tool. This awareness can help in learning to adjust compensation strategies, which may help to prevent musculoskeletal complaints in individuals with BPI.

Two studies evaluated the use of compensatory movements in individuals with BPI during activities of daily living and during specific movements of the BPI-affected upper limb (like reaching with the hand to the mount). Both studies used motion capture systems and showed that the movement patterns were task-specific and that individuals with BPI compensated mostly with their trunk and shoulder [2–4]. However, motion capture systems have challenges, including a) preparation time required for placing markers, b) lack of portability from the use of fixed cameras, or lengthy set-up requirements for portable systems, and c) equipment costs and training requirements [18]. Additionally, the equipment is costly and needs specialized staff. With an observational rating system, compensatory movements of individuals with BPI may also be measured reliably during specific tasks. Such a rating system does not have the disadvantages of motion capture systems.

To the best of our knowledge an observational rating system rating movement applicable for the use in individuals with BPI does not exist. Previously a qualitative rating system for rating shoulder and trunk movements in upper limb prosthesis wearers during the performance of FCE-OH tests was developed [19]. However, we expect that this rating system is not applicable for the use in individuals with BPI, because upper limb prosthesis wearers compensate for the loss of wrist, forearm (pro- and supination) and sometimes elbow movements due to the limited degrees of freedom of the prosthesis [20]. In contrast, individuals with BPI need to compensate for loss of sensation and strength resulting in a limited active range of motion. This probably will result in the use of different compensatory strategies. We, therefore, aimed (1) to develop and to explore the feasibility of an observational rating system for rating posture and movements of the shoulders and trunk in individuals with BPI during the performance of FCE-OH tests, (2) to explore the interrater and intrarater reliability of this new rating system.

## Methods

### Design

This study consisted of a development phase and a qualitative evaluation phase (Fig 1).

### Participants

Three groups of participants were included: raters, individuals with BPI and able-bodied controls. Video recordings of individuals with BPI and able-bodied controls, executing FCE-OH tests as part of previous studies, were used; all individuals signed informed consent before entering the previous study and gave consent to reuse video recordings for research purposes [16,21]. All raters signed informed consent prior to study entrance. The medical ethics committee of the University Medical Center Groningen decided that no formal approval was needed (METC file number: METc 2020/518).

**Development phase: planning**
- Define goal: a scoring system for rating postures and shoulder and trunk movements in individuals with BPI
- List movement patterns of individuals with BPI and control persons, and define movement patterns and nWNL postures
- nWNL movements and postures were: 1. ROM in a direction + 2SD and/or 2. additional movements compared to controls.
- Check if the system for rating compensatory movements in upper limb prosthesis wearers is applicable [19]

**Development phase: Construction**
- Construct the first draft of the scoring system (8 items)
- Perform pilot tests (1, 2, 3, 4, 5) until kappa of all items =0.6
  - Pilot tests 1, 2: hand therapists and physiotherapists without FCE-experience (n=2 per pilot test, per item 5 20-second video fragments were rated).
  - Pilot tests 3, 4 and 5: (allied) medical professionals without FCE-experience (n=5 per pilot test, per item 8 20-second video fragments were rated)
- After each construction phase pilot test items with ?<0.6 were adjusted based on the raters feedback

**Qualitative evaluation phase**
- Reliability testing: (inter) national FCE-experts (n=10), performed 2 rating sessions 2 weeks apart
  - Per rating session, per item 8 20-second video fragments were rated
- Feasibility assessed with a questionnaire

**Fig 1. Flowchart of the development and exploration of the reliability of the scoring system for rating postures and movement patterns in individuals with BPI during the performance of FCE-one handed tests.** *Abbreviations:* BPI, brachial plexus injury; ROM, range of motion; SD, standard deviation; FCE, functional capacity evaluation; κ, Fleiss kappa; nWNL, not within normal limits.

All procedures were in accordance with the ethical standards of the responsible committee on human experimentation (institutional and national) and with the Helsinki Declaration of 1975, as revised in 2000.

**Raters.** Development phase: all raters had a (allied) medical background and were experienced in observing movement patterns. All raters worked in the local medical center. Most raters had no experience with FCE tests. Five pilot tests were performed in this phase. In order to avoid a learning effect, all raters participated in one pilot test only. In the first and second pilot test a total of four hand therapists participated, two therapists per test. In the third pilot test five raters participated, one hand therapist and four physiotherapists. In the fourth and fifth pilot test a total of ten residents in physical rehabilitation medicine participated, five raters per test.

Qualitative evaluation phase: Twenty raters were recruited using an international Linked-In FCE research network and by contacting medical professionals at our local medical center. The sample size was based on two previous studies that assessed the reliability of an observational rating system [19,22]. All raters had a medical background and had experience in observing movement patterns. Experience with FCEs was desirable but not required. All raters had good understanding of Dutch or English. Recruitment of the raters took place from 19 November 2021 to 1 February 2022.

**Individuals with BPI.** Video recordings of 15 individuals with BPI performing FCE-OH tests, recorded during a previous study that aimed to determine the functional capacity of individuals with BPI [21], were reused. The characteristics of the individuals with BPI were described in Table 1. In the previous study, a physical examination was performed in order to assess the remaining function of the BPI-affected upper limb. The physical examination included an assessment of the active range of motion, strength, and sensation of the hand, see supporting information S1 Table. In order to protect the participants' privacy their faces were blurred. Camera positions were standardized (Table 2), the videotapes were recorded from March 2016 to June 2019 at the local medical center. Individuals with BPI who were between 18 and 65 years of age, who had been diagnosed with BPI, had remaining hand activity, had sufficient understanding of the Dutch language, performed paid work, and who had normal hand function in the sound hand were eligible to participate in the study. Exclusion criteria were hypertension (blood pressure >160/100 mmHg in rest), serious pulmonary conditions, cardiac conditions, or other conditions that could cause unsafe situations during physical effort exerted during test performance. The seven-item physical activity readiness questionnaire (PAR-Q) was used to screen participants for serious health problems [23]. Participants responding "Yes" to one or more of the items were excluded.

**Table 1. Characteristics of individuals with BPI and controls.**

| | Individuals with BPI (n = 15) | Controls (n = 21) |
|---|---|---|
| Gender, male | 13 (87) | 18 (86) |
| Mean age (years) | 49.9 ± 10.9 | 45.8 ± 11.7 |
| Mean time since onset BPI (years) | 9.6 ± 9.7 | N/A |
| Remaining active range of motion of the BPI affected arm. | • Full active range of motion: 4 (27)<br>• Diminished active range of motion in at least one joint: 11 (73) | N/A |

Data is presented as n(%) or mean ± SD.

Abbreviations: BPI, brachial plexus injury; SD, standard deviation; N/A, not applicable.

**Table 2. FCE-OH tests, participant instructions, FCE-outcome, camera positions and video fragment selected for rating.**

| FCE-test | Instructions | FCE-outcome | Camera position | Selected video fragment |
|---|---|---|---|---|
| Overhead lifting test two handed | Lift the plastic receptacle containing weights from table height to crown height 5 times within 90 seconds. After every 5 lifts the weight is increased till the maximum weight in reached. | The maximum lifted weight (kg) | Behind the participant | The second attempt of 5 lifts |
| Overhead working test | Stand with the hands on crown height with a cuff weight of 1 kg around the wrist of the unaffected side. Manipulate nuts and bolts. Hold this position as long as possible | Time this position is held (sec) | In front of the participant | The final 20 seconds |
| Overhead lifting test one handed | Lift a weight of 1.9 Kg 20 times from table height to crown height with one hand | Time needed for 20 lifts (sec) | Behind the participant | The final 20 seconds of the task performance with the affected side |
| Repetitive reaching test | Sit between two clicking systems on wingspan and alternate clicking each button for a total of 30 times as fast as possible | Time needed to press each button 30 times (sec) | In front of the participant | The final 20 seconds of the task performance with the affected side |
| Fingertip dexterity test | Sit in front of the pegboard (Purdue pegboard, model 32020 J.A. Preston Corporation New York, NY, USA) and place the pins in the board as fast as possible | The number of pins placed in the pegboard in 30 seconds | In front of the participant | The final 20 seconds of the task performance with the affected side |

Abbreviations: kg, kilogram; sec, seconds.



**Able-bodied controls.** Video recordings of 21 able-bodied controls (Table 1) performing FCE-OH tests at the local medical center in 2014, made during a previous study [16], were reused. Inclusion criteria for controls were: aged between 18 and 67 years, good understanding of Dutch or English, and normal hand function in both hands. Performing paid work was not an inclusion criterion for controls; the inclusion criterion was added for individuals with BPI because of a previous study that compared functional capacity to physical work demands [24]. Exclusion criteria for controls were similar to those for individuals with BPI. Controls were aware of the study and knew they were participating as controls.

## FCE-OH tests

Five out of six FCE-OH tests were selected: the overhead lifting test two-handed, the overhead working test, the repetitive reaching test, the fingertip dexterity test and the overhead lifting test-one handed. The hand grip strength test was not selected, because movements of the trunk and shoulders were not allowed in this test (Table 2).

## Scale development

The first 3 out of 4 phases for instrument development and validation were followed [25]: planning, construction, qualitative evaluation and validation. Fig 1 shows the steps taken in each phase.

**Planning phase.** In this phase, deviating movement patterns of the shoulders and trunk of individuals with BPI were identified. After the identification of deviating movements and postures, it was checked if these movements and postures could be rated with the previously developed rating system for upper limb prosthesis wearers, or that a new rating system specifically for individuals with BPI needed to be developed.

Movement patterns of the shoulders and trunk of individuals with BPI were compared to the controls for each FCE-OH test separately, because compensatory movements seem to be task specific [3]. Postures and movements deviating from controls were defined as not within normal limits (nWNL), if (1) the maximum range of motion deviated more than two standard deviations from the mean of the control group or if (2) additional movements were performed that were not observed in the control group. These criteria were based on the previously developed observational rating system for upper limb prosthesis wearers [19]. The range of motion in a direction was measured using VideoStudio Pro 2020 Corel corporation, allowing to play videos frame by frame. The frames in which the range of motion was maximal were chosen and angles were manually measured. In order to prevent missing deviating movements and postures, all videotapes of individuals with BPI were analysed separately and the range of motion of each individual was compared with the mean range of motion of controls for each test. One medical student listed the deviating postures and movements, the listed movements were checked by a medical professional. In case the medical professional disagreed with the medical student video recordings were reassessed together. The findings of the planning phase were discussed in the research group (which consisted of 3 rehabilitation physicians and one FCE-expert) until consensus was met. Only three out of eight items of the rating system for upper limb prosthesis wearers were suitable for use in individuals with BPI. These items included (1) trunk and (2) shoulder movements from the two-handed overhead lifting test, as well as (3) trunk movements from the fingertip dexterity test [11]. Therefore, a new rating system needed to be developed for individuals with BPI.

**Construction phase.** The first draft of the new rating system consisted of the three items that matched the observed movement patterns supplemented with five adjusted items (S1 Fig). Based on our previous experiences with developing a comparable rating system in upper limb prosthesis wearers, we decided to rate all items dichotomously (within normal limits (WNL) or (nWNL) [19].

The first draft was pilot-tested five times and adjusted according to the feedback of the raters (Fig 1). All pilot tests were performed online, guided by a moderator (TMJL) using Microsoft Teams. Before rating an item, raters received a short instruction and four example video fragments were shown (two test performances WNL and two test performances nWNL). After rating all items, the raters were asked to provide feedback on the instructions and the rating system. Some video recordings appeared to be more difficult to rate than others. In order to reduce the influence of one single video



recording on the results, all raters participating in the pilot test three, four, and five rated eight video fragments of individuals with BPI performing the FCE-OH tests. Raters of these pilot tests were also asked to rate the certainty of their ratings (dichotomous, certain: yes or no), in order to identify difficulties in rating.

**Qualitative evaluation phase.** In this final phase interrater and intrarater reliability were explored (Fig 1). Each rater was asked to participate in two rating sessions, with two weeks between rating sessions. Prior to each rating session the use of the rating system was explained by an instruction video, containing two examples of a performance WNL and two performances nWNL of each FCE-OH test. For each item, except for the overhead lifting test one-handed, the same eight video fragments as in the pilot tests were selected for rating. For the overhead lifting test one-handed seven video fragments were selected for rating, because only seven individuals with BPI were able to perform this test with the affected upper limb. The video instruction and all video fragments were offered online, using a secured video fragment rating system (VFR MAS Outreach, Leeuwarden the Netherlands). This system allows to share videos, without the possibility to download them, share the videos to others or to distribute them in any other way. Raters were instructed to rate independently and not to replay or pause the videos during rating in order to mimic a clinical setting as best as possible. As in the pilot tests, raters were asked if they were certain of their ratings. After each rating session raters handed in their results and were instructed to remove their rating form from their computer. After the second rating session raters were asked to provide (qualitative) feedback on the feasibility and usability of the rating system and the video instruction, using a self-developed questionnaire (S1 File). This questionnaire had previously been used during the development of the rating system for upper limb prosthesis wearers, but has not been formally validated. Rating took place from January to March 2022.

## Statistical analysis

Construction phase: interrater reliability for all items in the pilot tests was determined using Fleiss Kappa (κ) for multiple raters. Qualitative evaluation phase: Fleiss kappa for multiple raters was used to determine the interrater reliability, Cohen's kappa was used to determine intrarater reliability. Because of categorical data, kappa statistics were used to determine the interrater and intrarater reliability [26,27]. Percentages of absolute agreement and 95% confidence intervals (CI) were determined. Reliability was considered poor if $κ ≤ 0.4$, moderate if $0.41 < κ < 0.59$, sufficient if $0.6 ≤ κ < 0.79$ and good if $κ ≥ 0.80$ [28]. In case of missing data the ratings of that particular rater were omitted from the analyses for the interrater reliability. For the intrarater reliability analysis, corresponding ratings of the first and second rating session were removed pairwise in case of missing data. Post-hoc Gwets AC 1 was determined to assess inter- and intrarater reliability, because of the risk of prevalence bias when using kappa statistics [28,29]. The feedback provided by the raters through the questionnaire was analyzed using descriptive statistics only. Statistical analyses were performed using Statistical Package for the Social Sciences (SPSS) version 25.0 software package (SPSS; IBM Corp, Armonk, NY). AgreeStat 360 was used to calculate Gwets AC1 [30].

## Results

### Construction phase

The results of the construction phase are shown in S1 Fig. The most important adjustment after the five pilot tests was that the descriptions were focused more on symmetry, in order to improve the rating of subtle deviating movements such as shoulder elevation in both two-handed FCE-OH tests. We did not develop separate items for these subtle deviating movements, because from the development of the previous observational rating system for upper limb prosthesis wearers, we learned that these items were difficult to rate reliably by observation [19]. The final draft consisted of five items, one item per FCE-OH test (see also S2 File). Points of attention were specified per item. Raters were instructed to rate the worst performance in case of a variable movement pattern. Fig 2 shows an illustrative example of the overhead lifting test two-handed item.

POINTS OF ATTENTION

- Rate the <u>worst</u> performance
- Pay attention to symmetrical movement of the glenohumeral joints and elbows during both the upward and downward movement.

| | Example of symmetry | Example of asymmetry |
|---|---|---|
| | | |
| | Symmetry or minimal asymmetry (0 points) | Clearly visible asymmetry (1 point) |
| | Glenohumeral joints and elbows move approximately symmetrical during the whole task. | Clearly visible asymmetrical movements of glenohumeral joints and/or elbows. |
| Score | | |
| | | |

**Fig 2. Illustrative example of the overhead lifting test two-handed item in the final draft of the scoring system.**

## Qualitative evaluation phase

In total 20 raters were recruited, eight from Austria and twelve from the Netherlands. The first rating session was performed by 17 raters (5 males, 12 females; professions: 8 physiotherapists, 5 rehabilitation physicians, 2 occupational therapists and 2 residents for rehabilitation medicine), while 16 out of these 17 raters completed the second rating session as well. Two raters had time constraints and two raters gave no reason for not performing one or both rating sessions. Seven raters had FCE-experience (mean years of FCE-experience $3.3 \pm 7.7$ years, missing data $n = 1$). The mean duration of work experience of raters without FCE-experience was $10.7 \pm 11.8$ years (missing data $n = 2$). The mean time between the first and the second rating session was $16.3 \pm 3.8$ days. Raters were able to open all documents and watch the videos, one rater had problems because of stuttering videos. Almost all raters declared that the instructions for the testing procedure were clear. One rater answered the instructions were unclear, but stated a lack of clarity in the rating system as the reason and not in the instructions. A post-hoc analysis to test whether the results were influenced by this rater showed similar results for analyses that included or excluded this rater. The rating sessions were conducted between 17 January 2022 until 2 March 2022.

The overall interrater reliability of both rating sessions was moderate (Table 3). Fleiss kappa increased in the second rating session and was almost sufficient. In both rating sessions 77% of the raters were certain of their ratings. In the second rating session the interrater reliability of the overhead working test was good and sufficient for the repetitive reaching test and the fingertip dexterity test. The interrater reliability of both overhead lifting tests was poor in both rating sessions. Most raters (19−30%) mentioned that they were uncertain about their ratings for both overhead lifting tests. The interrater reliability was similar for raters with and without FCE experience (first rating session raters with FCE-experience $\kappa = 0.56$, 95% CI 0.41–0.70, raters without FCE-experience $\kappa = 0.53$, 95% CI 0.39–0.66; second rating session raters with FCE-experience $\kappa = 0.64$, 95% CI 0.56–0.73, raters without FCE-experience $\kappa = 0.58$, 95% CI 0.43–0.73).

**Table 3. Inter and intrarater reliability of the observation-based scoring system.**

| FCE-OH test | Interrater reliability 1st session | | | | Interrater reliability 2nd session* (n=16) | | | | Intrarater reliability (n=16) | | |
|---|---|---|---|---|---|---|---|---|---|---|---|
| | % agree-ment | κ (95% CI) | AC1 (95% CI) | Certain mean n(%)$ | % agree-ment | κ (95% CI) | AC1 (95% CI) | Certain mean n(%)$ | % agree-ment | κ (95% CI) | AC1 (95% CI) |
| Overhead work-ing test | 88 | 0.75 (0.52-0.99) | 0.76 (0.73-1.00) | 15.0 (88) | 94 | 0.88 (0.75-1.00) | 0.89 (0.78-1.00) | 14.2 (89) | 91 | 0.80 (0.69-0.91) | 0.83 (0.72-0.93) |
| Repetitive reaching test | 78 | 0.56 (0.28-0.84) | 0.58 (0.22-0.94) | 12.3 (72) | 81 | 0.62 (0.30-0.94) | 0.64 (0.23-1.00) | 12.1 (76) | 84 | 0.67 (0.54-0.80) | 0.70 (0.57-0.83) |
| Fingertip dex-terity test | 77 | 0.55 (0.32-0.77) | 0.55 (0.31-0.79) | 13.9 (82) | 85 | 0.69 (0.44-0.94) | 0.72 (0.43-1.00) | 11.1 (79) | 82 | 0.63 (0.49-0.76) | 0.64 (0.50-0.78) |
| Overhead lifting test 2-handed | 62 | 0.25 (0.00-0.50) | 0.25 (−0−10-0.61) | 12.0 (71) | 64 | 0.28 (−0.02-0.57) | 0.28(−0.09-0.65) | 9.8 (61) | 81 | 0.62 (0.49-0.76) | 0.63 (0.48-0.76) |
| Overhead lifting test one- handed | 65 | 0.29 (0.05-0.54) | 0.30 (−0.01-0.60) | 14.4 (73) | 70 | 0.40 (0.07-0.73) | 0.42 (0.01-0.84) | 12.0 (80) | 70 | 0.41 (0.24-0.58) | 0.42 (0.24-0.60) |
| Overall | 74 | 0.48 (0.36-0.60) | 0.49 (0.36-0.62) | 13.1 (77) | 79 | 0.59 (0.45-0.72) | 0.58 (0.44-0.72) | 11.9 (77) | 82 | 0.64 (0.58-0.70) | 0.64 (0.58-0.71) |

Missing data: Fingertip dexterity test (second rating session): 2 missing data; Overhead lifting test one-handed (second rating session): 7 missing data (see also S2 Table).

$Mean number of raters that answered to be certain about their scores on the rated video fragments.

* Because of missing data, interrater reliability for the second session was determined using the scores of 14 raters for the fingertip dexterity test and the scores of 15 raters for the overhead lifting test one-handed.

Abbreviations: FCE-OH test, functional capacity evaluation one-handed test; κ, Fleiss kappa; % agreement: percentage of absolute agreement; 1st session, first rating session; 2nd session, second rating session; 95% CI, 95% confidence interval; n, number of raters; AC1, Gwet's AC1.

The overall intrarater reliability was sufficient. The intrarater reliability was good for the overhead working test and sufficient for the other items except for the overhead lifting test one-handed. The overall intrarater reliability was similar for raters with and without FCE-experience (raters with FCE-experience κ=0.64, 95% CI 0.58–0.70, raters without FCE-experience κ=0.68, 95% CI 0.61–0.76).

Post hoc determined Gwets AC1 values were similar to kappa values, indicating no prevalence biases (Table 3).

The intra- and interrater reliability may have been influenced by the insufficient power of the kappa analyses. Therefore we conducted a post hoc power analysis in R (version 4.3.1; R Core Team) using custom-made scripts developed in RStudio and a web-based sample size calculator [31]. These analyses indicated that the sample size was sufficient for the overall intrarater reliability (number of videos required 10, for a power of 0.8 with kappa expected 0.6, precision 0.1 and number of raters 16) and interrater reliability (246 rating required, for a power of 0.8 with kappa expected 0.6, precision 0.1). However, the intra- and interrater reliability analyses for the individual items were underpowered.

## Feasibility and usability

Fifteen raters provided feedback on the feasibility and usability of the developed rating system (Table 4). In their feedback raters complained that the camera position was not perfect in the frontal plane in some videotapes. Furthermore, they suggested adding feedback to the video instruction to improve the training.

## Discussion

A new system was developed for rating deviant postures and movements of the shoulders and trunk in individuals with BPI during the performance of FCE-OH tests. The interrater and intrarater reliability were explored. The overall interrater

**Table 4. Feedback of the raters on the developed rating system.**

| Statement | Raters (n = 15) |
|---|---|
| The rating system is easy in use (agree) | 14 (93), missing data 1 |
| The scoring system is easy to use in daily practice. | Totally agree: 6 (30) |
| | Agree: 2 (10) |
| | Neutral: 3 (15) |
| | Disagree: 3 (15) |
| | Totally disagree: 1 (5) |
| Watching the instruction video is sufficient training to use the scoring system in daily practice. | Totally agree: 2 (10) |
| | Agree: 4 (20) |
| | Neutral: 3 (15) |
| | Disagree: 4 (20) |
| | Totally disagree: 1 (5) |
| | Missing data 1 (5) |

Data presented as n (%).

reliability of the rating system was moderate in both rating sessions, although nearly sufficient in the second rating session. The overall intrarater reliability was sufficient.

The developed rating system was based on the rating system for compensatory trunk and shoulder movements in upper limb prosthesis wearers [19], this rating system appeared not to be applicable for the use in individuals with BPI. In both rating systems the interrater reliability was higher compared with the interrater reliability. The interrater reliability of the rating system for individuals with BPI was lower compared with the rating system developed for upper limb prosthesis wearers. Rating movement patterns in individuals with BPI appeared to be more challenging, because of greater variation in remaining upper limb function compared with upper limb prosthesis wearers, resulting in greater variability in movement patterns. This variability was also observed in children with brachial plexus birth injury performing modified Mallet scale tasks [4]. The variability in movement patterns may have caused difficulties in recognizing and rating movement patterns and postures, because the deviant movement patterns in the video fragments selected for rating were sometimes different from the examples of deviant movement patterns shown in the video instructions. This issue was particularly apparent in the two overhead lifting tests, where a combination of movements had to be rated simultaneously. Pilot testing had indicated that scoring these components individually was problematic (see also supportive information S1 Fig), which led to the decision to assess them as combined movement patterns. However, the substantial variation in how these combined movements were performed likely increased the complexity of the rating and may therefore explain the low interrater reliability observed for the overhead lifting tests.

The interrater and intrarater reliability of the rating system was similar for medical professionals with and without FCE-experience, suggesting that specific FCE-experience is not required for the use of the rating system after following a training program. The interrater reliability increased from the first to the second rating session, which was also observed in other studies [19,22]. This finding may suggest a learning effect; however, other unexamined rater specific factors, such as fatigue, mental health or time taken for rating, may also have contributed.

The combination of sufficient intrarater reliability and insufficient reliability may have been a consequence of the limited rating training. Only 40% of the raters agreed that the video instruction was sufficient before the use of the system. Furthermore, when they handed in their ratings, some raters mentioned that they would have liked to receive feedback on their ratings in order to know if their ratings were right. Clearly, a single video instruction should be considered insufficient for future use. A more extensive training program that incorporates structured feedback may help to clarify rating criteria

and help raters to recognize and rate variable movement patters of individuals with BPI and thereby improve interrater reliability.

We developed an observational rating system for all individuals with BPI. Observations may also improve by developing different rating systems for different levels of BPI. Probably the variation in movement patterns will decrease when rating individuals with similar levels of BPI, which could make it easier to recognize movement patterns and thereby simplify rating. The disadvantage of different systems is that raters need to train and maintain rating skills for different rating systems. This may be challenging, especially because of the low prevalence of BPI [32,33]. It may nevertheless be necessary to opt for optoelectrical motion capture systems for reliable measurement of deviant movements and postures in individuals with BPI, despite the disadvantages of these systems [18].

It was remarkable that almost all raters agreed with the statement that the system was easy in use, but that only 53% of the raters agreed with the statement that the system would be easy in use in clinical practice. This finding is in contrast to the rating system for compensatory trunk and shoulder movements in upper limb prosthesis wearers, where all raters agreed with the statements that the system was easy in use and could be easily implemented in clinical practice [19]. Unfortunately, raters were not asked why they did not agreed with the statement that the system would be easy to use in clinical practice. Insufficient training may be one reason. Another reason may be that raters experienced rating as difficult, indicated by uncertainty of their ratings, which may explain the disagreement with the statement that the system is easy in use in clinical practice. This finding needs further investigation.

Based on the reliability test results and the comments of the raters, the current developed rating system cannot be used in clinical practice yet. It can be questioned if an observational ratings system is the right measurement tool to measure deviating movement patterns and postures in individuals with BPI, because of the variability in movement patterns. Follow-up research should therefore focus on the use of an optoelectrical movement system to measure movement patterns and postures during the performance of FCE-OH tests, to provide more insight into the qualitative aspects of the FCE-OH test performances in individuals with BPI.

## Limitations

It is plausible that certain deviating movement patterns and postures remain undetected by the developed rating system. This limitation may be caused by the definition of movement patterns classified as nWNL, which was based on a maximum range of motion deviating by more than two standard deviations from the mean of the control group. Consequently, this definition may have caused that smaller deviating movements were not identified as nWNL. However, from a clinical perspective the more severe deviating movements may be the most important to target in order to prevent MSCs. These movements cause larger internal moments around joints and thus require more muscle force, leading to muscle fatigue, which subsequently may increase the risk on MSCs [5].

Raters reported that the use of fixed camera positions in a single plane caused difficulties during rating. Additionally the fixed camera positions may have caused that certain movement patterns and postures in other planes were not observed during the planning phase. Therefore the use of multi angle recordings is desired in future studies.

Deviating movements and postures of individuals with BPI were determined by manually measuring the range of motion from video frames by two researchers. This approach primarily captured deviations large enough to be assessed by observation; however, subtle deviations may have gone undetected. The validity and reliability of this method are unknown, and therefore results of the planning phase should be interpreted with caution. For future research, the use of motion capture systems should be considered to determine deviating movement patterns and postures. Previous studies have demonstrated that such systems also identify smaller deviation movement patterns of individuals with BPI, like scapulothoracic movements, rotation of the humerus and movement of the scapula and clavicle with respect to the thorax (virtual thoracohumeral movement) [2,4,34].

The item selection in the planning phase was based on an internal consensus group consisting of three medical professionals and one FCE-expert. In future research, the validity of the rating system may be enhanced by performing a Delphi study, as it systematically incorporates expert consensus and reduces individual bias.

The sample size of individuals with BPI was small. Post hoc we checked if saturation of the observed movement patterns and postures was reached for each FCE-OH test. Saturation was reached if no new deviating movement patterns and postures were observed in at least the last four analyzed videotapes. In four out of five FCE-OH tests saturation of the observed deviating movement patterns and postures was reached. Only for the overhead lifting test, no saturation was reached.

A post-hoc power analysis showed that the power of the overall interrater and intrarater reliably analyses was sufficient, however the power of reliability analyses of the individual items was insufficient. A lower number of raters, rating more videotapes per item would have increased the power [28]. However, this would lead to long-lasting rating sessions which was undesirable because it could lead to fatigue and time constraints for the raters, which in turn could negatively affect reliability. Therefore and because of the explorative character of this study, it was decided to increase the number of raters instead of the number of videotapes that needed to be rated.

## Conclusions

A rating system was developed to rate movement patterns and postures of the trunk and shoulders in individuals with BPI during the performance of FCE-OH tests by observation. Movement patterns and postures of individuals with BPI could be rated reliably by the same rater using the developed rating system in combination with the video instruction. The interrater reliability was moderate in the first rating session and almost sufficient in the second rating session (κ was 0.59, whereas κ ≥ 0.6 was considered sufficient). The interrater reliability of the items of both overhead lifting tests was low in particular. The moderate interrater reliability of the rating system may be induced by the great variation in movement patterns in individuals with BPI, caused by differences in the remaining function of the affected upper limb. No differences were observed in the results of raters with or without FCE-experience, which implies that FCE-experience is not required for the use of the system. The feasibility of the rating system could be improved by implementing a more extended training program. Further research is needed to determine how the reliability of measuring deviant movement patterns and postures in individuals with BPI can be increased, taking advantages and disadvantages for use in daily practice into account.

## Supporting information

**S1 Table. Physical examination: Assessed range of movement and muscles selected for strength testing.**
(DOCX)

**S2 Table. Results of the qualitative evaluation phase.**
(DOCX)

**S1 Fig. Flow chart of the construction phase of the development of the scoring system for rating postures and movement patterns in individuals with BPI during the performance of FCE-one handed tests.**
(DOCX)

**S1 File. Questionnaire qualitative evaluation phase.**
(DOCX)

**S2 File. Qualitative scoring system for rating posture and movements of the shoulders and trunk in individuals with brachial plexus injury during the performance Functional capacity evaluation one-handed (FCE-OH)\*.**
(DOCX)



## Acknowledgments

The authors thank M. Reijmerink for his help in listing deviating postures and movement patterns.

Furthermore we would thank all raters who participated in this study.

## Author contributions

**Conceptualization:** Tallie M. J. van der Laan, Sietke G. Postema, Corry K. van der Sluis, Michiel F. Reneman.

**Data curation:** Tallie M. J. van der Laan, Michiel F. Reneman.

**Formal analysis:** Tallie M. J. van der Laan, Michiel F. Reneman.

**Investigation:** Tallie M. J. van der Laan, Sietke G. Postema, Corry K. van der Sluis, Michiel F. Reneman.

**Methodology:** Tallie M. J. van der Laan, Sietke G. Postema, Corry K. van der Sluis, Michiel F. Reneman.

**Project administration:** Sietke G. Postema, Corry K. van der Sluis, Michiel F. Reneman.

**Supervision:** Sietke G. Postema, Corry K. van der Sluis, Michiel F. Reneman.

**Writing – original draft:** Tallie M. J. van der Laan.

**Writing – review & editing:** Tallie M. J. van der Laan, Sietke G. Postema, Corry K. van der Sluis, Michiel F. Reneman.

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
