## [Decision Letter · Decision Letter 0]

1 Oct 2025

Dear Dr. van der Laan,

Thank you for submitting your manuscript to PLOS ONE. After careful consideration, we feel that it has merit but does not fully meet PLOS ONE’s publication criteria as it currently stands. Therefore, we invite you to submit a revised version of the manuscript that addresses the points raised during the review process.

We look forward to receiving your revised manuscript.

Kind regards,

Priti Chaudhary, M.S.

Academic Editor

PLOS ONE

Journal Requirements:

3. We note that your Data Availability Statement is currently as follows: “All relevant data are within the manuscript and its Supporting Information files.”

Additional Editor Comments:

**Authors are required to reply all the queries, raised by the reviewers.**

Reviewer's Responses to Questions

**Comments to the Author**

1. Is the manuscript technically sound, and do the data support the conclusions?

Reviewer #1: Yes

Reviewer #2: Partly

Reviewer #3: Yes

2. Has the statistical analysis been performed appropriately and rigorously?

Reviewer #1: N/A

Reviewer #2: Yes

Reviewer #3: Yes

3. Have the authors made all data underlying the findings in their manuscript fully available?

Reviewer #1: No

Reviewer #2: Yes

Reviewer #3: Yes

4. Is the manuscript presented in an intelligible fashion and written in standard English?

Reviewer #1: Yes

Reviewer #2: No

Reviewer #3: Yes

Reviewer #1: Reviewer Report

General Assessment and Recommendation

This manuscript addresses an important gap in the evaluation of compensatory movement patterns in individuals with brachial plexus injury (BPI) during one-handed functional capacity evaluation (FCE-OH). The authors propose and test a novel observational scoring system for identifying compensations, and report inter- and intra-rater reliability based on video assessments by 16 raters. The study is original, methodologically sound in many respects, and clinically relevant. However, several methodological, statistical, and reporting issues need clarification and strengthening. I therefore recommend Major Revision.

1. Scope and Novelty

The manuscript is well-positioned: FCE-OH is widely used for work capacity evaluations, yet does not capture how tasks are performed. The observational scoring system adds a qualitative layer that can be highly valuable in preventing musculoskeletal complications due to compensatory movements.

Prior tools developed for prosthesis users are not directly transferable, which justifies the development of a new scale specifically for BPI.

The work is timely and relevant to rehabilitation, occupational health, and clinical ergonomics.

Strengths: International pool of raters, two test sessions to assess stability, item-level analysis, and the addition of an applicability survey.

2. Study Design and Scale Development

Major Revision Request 2.1 – Content validity:

The development process is described but lacks systematic reporting of item generation and expert consensus. How were “within normal limits (WNL)” versus “not WNL” decisions operationalized? The current items are largely symmetry-based; clearer operational definitions and anchor examples (possibly visual diagrams) should be provided in the supplementary material.

Major Revision Request 2.2 – Stratification:

Given the heterogeneity of BPI, stratification by residual function (e.g., ROM, strength categories, lesion levels) is warranted to illustrate item performance across subgroups. This would help clarify whether some items are only reliable in particular clinical subpopulations.

Major Revision Request 2.3 – Training protocol:

Over half of the raters felt that the video-based training was insufficient. A structured training module (e.g., a set of “gold standard” clips with feedback and short quiz) should be outlined, and the improvement from session 1 to session 2 should be explicitly analyzed as a “learning effect.”

3. Statistical Analysis and Reliability

Major Revision Request 3.1 – Kappa limitations:

Low kappa values despite moderate absolute agreement suggest prevalence/bias effects. The authors should include prevalence and bias indices, and/or alternative coefficients such as Gwet’s AC1/AC2 or PABAK.

Major Revision Request 3.2 – Advanced modeling:

If feasible, consider multi-level approaches (e.g., Generalizability Theory or mixed-effects logistic regression), which can simultaneously model variance due to rater, video, and session factors.

Major Revision Request 3.3 – Item-level behavior:

The weakest performance was for overhead lifting items. These may conflate several compensations (shoulder, trunk, elbow). Splitting them into subcomponents could improve clarity and rater agreement.

Revision Request 3.4 – Power and CI width:

Confidence intervals are relatively wide. While the exploratory nature is acknowledged, a brief power calculation or justification of sample size relative to target kappa would add transparency.

4. Video Protocol and Practical Implementation

Several raters reported difficulty due to loss of the sagittal plane in some videos. For future work, multi-angle recordings (front, side, posterior) or camera tracking should be considered.

The instruction that raters could not pause or replay videos mirrors real clinical practice, but should be explicitly justified and included in the user manual.

5. Interpretation of Results

The improvement from session 1 to session 2 suggests a training effect. Quantifying this difference statistically (e.g., confidence intervals for session-to-session change in kappa) would strengthen the claim.

The interpretation of “almost sufficient” reliability should be replaced with numeric ranges and standard thresholds for clarity.

The finding that intra-rater reliability is sufficient while inter-rater reliability is moderate should be highlighted as both a limitation and an opportunity for further rater training.

6. Data Availability and Ethics

Ethics approval and informed consent are clearly stated.

Data availability statement is appropriate. However, I recommend sharing the raw rating matrix, anonymized survey results, rater instructions, and—if ethically permissible—short example video clips in an open repository with DOI. This will greatly facilitate replication and training.

7. Writing, Language, and Presentation

Minor typographical errors should be corrected (e.g., “abled bodied” → “able-bodied,” “prothesis” → “prosthesis”).

Abbreviations should be defined upon first use (e.g., FCE-OH, WNL).

Tables and figures should be polished: abbreviations expanded, statistical measures (κ, CI, % agreement, prevalence) clearly presented.

Supplementary materials should include the final scoring system in a single consolidated PDF.

Reviewer #2: 1. Claims & Framing

• Abstract results, discussion and conclusions overstate κ values as “sufficient,” whereas they are best described as moderate (κ≈0.59–0.64).

• The novelty of the tool compared with prior prosthesis-user and observational/postural tools is not sharply positioned in discussion and introduction.

2. Scale Development (Methods)

• The reliance on one student, one professional, and internal group consensus limits the robustness of the item selection; broader expert involvement (e.g., multiple clinicians or a Delphi panel) would improve validity.

• The choice of a >2 SD threshold is statistical but not clearly justified clinically. Why is this cutoff appropriate for defining abnormal compensation?

• The binary WNL/nWNL scoring system is simple and practical, but may oversimplify nuanced compensatory behaviors. A graded scale (e.g., mild/moderate/severe deviation) could capture variation more effectively.

3. Raters, Training, and Blinding (Methods)

• The manuscript does not describe how raters were trained, calibrated, or whether their backgrounds (clinical vs non-clinical) might have influenced ratings. This is a crucial omission, as interrater reliability depends heavily on training consistency.

• It is unclear whether raters were blinded to diagnosis (BPI vs control). Knowledge of group assignment could bias assessments.

• The questionnaire is only briefly mentioned. No details are provided regarding its design, validation, or analysis. Without transparency, these findings cannot be meaningfully interpreted.

4. Sample Size & Power (Methods)

• The sample size (15 BPI, 21 controls, 16 raters) is modest. No power calculation or justification is provided. Reliability studies require such justification to ensure κ estimates are stable.

5. Statistics & Interpretation

• The Results section is statistically adequate but lacks depth: (1. Were reliability values consistent across different FCE tasks, or did some tasks show higher agreement? 2. Were there systematic differences in rating patterns between raters with different professional backgrounds? 3. Were there differences in reliability between the BPI and control groups?)

• The discussion should address why interrater reliability improved from session 1 to session 2. Was this due to learning effects, improved familiarity with the tool, or random variation? Without exploring this, the findings are difficult to contextualize.

• Although most raters described the tool as “easy to use,” only ~50% considered it clinically feasible, citing insufficient training. This contradiction requires deeper analysis.

6. Language & Structure

• Minor grammar/typography:

The keywords in the PLOS ONE submission system contain a typo and should be carefully checked.

Line 238: “video’s” should be corrected to “videos.”

Consistently use prosthesis (not prothesis).

Perform a thorough proofread to correct similar small grammar, pluralization, and capitalization issues throughout the text.

• Limitations section:

The Limitations are presented in great detail, but the section currently reads as overly long, repetitive, and weighted with procedural details. While transparency is commendable, the narrative would benefit from focusing on the most impactful issues and reframing them in a forward-looking way.

Reviewer #3: ⦁ Can you expand more on the background part on the abstract

⦁ In the introduction part, "Individuals with brachial plexus injury (BPI) need to compensate for the loss of function of theaffected arm." I suggest first to give a brief mechanism of action and intervention of the brachial plexus anatomy and physiology

⦁ Can you mention if the able bodied controls aware of the research and that they are being controls or if it was blinded to them?

⦁ In the mentioned raters part "Twenty raters were recruited using an international Linked-In FCE research network and by contacting medical professionals at our local medical center" can you mention more about the expertise of the raters whether they are MDs or experts in Orthopedics or Physical Therapy.

⦁ From this part "Recruitment of the raters took place from 19 November 2021 to 1 February 2022" can you mention also the time frame of observing movement patterns

⦁ in Line 148 "Video recordings of 15 individuals with BPI performing FCE-OH tests (2 females, 13 males, mean age 49.9 ± 10.9 years, mean time since onset BPI 9.6 ± 9.7 years, recorded during a previous study that aimed to determine the functional capacity of individuals with BPI" i suggest and encourage the authors to provide a table for better visualization and show data

⦁ Also in Line 153-154 "four had a full active range of motion of the affected limb and eleven had a diminished active range of motion of at least one joint caused by contractures or limited strength. also a table here will lead to a better visualization

⦁ Same Applies to line 169

⦁ Please reorganize wording and clarity in here "Only three out of eight items of the rating system for upper limb prosthesis wearers were suitable for use in individuals with BPI (overhead lifting test two211 handed; items trunk and shoulder movements and fingertip dexterity test; item trunk movements)" as which were the 3 items used

⦁ in Line 223 " Some video recordings appeared to be more difficult to rate than others. In order to reduce the influence of one single video recording on the results, eight video fragments were rated in pilot test 3, 4 and 5. Raters of these pilot tests were also asked to rate the certainty of their ratings (dichotomous, certain: yes or no), in order to identify difficulties in rating." Were single Videos was evaluated by one rater only? if yes can you mention that in the limitations section as operator bias might present

⦁ In Line 318, Feasibility and usability I would also encourage having a diagram, table, or chart presenting the data

⦁ Great Use of the matrix of discussion. a one point to add is just comparing current study findings to the two studies that evaluated the use of compensatory movements you mentioned before

⦁ I suggest restructuring the discussion and limitation parts as limitation part have an enormous amount of paragraphs suitable in the discussion

.

Reviewer #1: No

Reviewer #2: **Yes:**Dr.Fawwaz Al-SmadiDr.Fawwaz Al-SmadiDr.Fawwaz Al-SmadiDr.Fawwaz Al-Smadi

Reviewer #3: **Yes:**Nasser F. AlSunbulNasser F. AlSunbulNasser F. AlSunbulNasser F. AlSunbul

---

## [Author Response · Author response to Decision Letter 1]

22 Feb 2026

Dear reviewers,

Thank you for the helpful comments to improve our manuscript. Based on these comments we have modified the text and tables. Below we respond to your comments in detail.

Reviewer 1

Major Revision Request 2.1 – Content validity:

The development process is described but lacks systematic reporting of item generation and expert consensus. How were “within normal limits (WNL)” versus “not WNL” decisions operationalized?

• Response: Within normal limit was operationalized as: the maximum range of motion deviated maximal two standard deviations from the mean of the control group.

Not within normal limits was operationalized as: the maximum range of motion deviated more than two standard deviations from the mean of the control group or if additional movements were performed that were not observed in the control group, see page 11, line 217-220. The criteria for movements and postures that were classified as “not within normal limits” were based on the criteria we used to develop the ratings system for upper limb prothesis wearers [19]. We added this to the manuscript, page 11, line 220-221. We selected movements and postures that deviated more than 2 standard deviations from the mean range of motion measured in the control group, because with this criterion 95%of the normal movement patterns should be within the selected range. With this criterion we operationally defined movement patterns classified as not within normal limits.

The current items are largely symmetry-based; clearer operational definitions and anchor examples (possibly visual diagrams) should be provided in the supplementary material.

• Response: Only the two two-handed FCE-OH tests were assessed based on symmetry; in these FCE-OH tests evident asymmetry was not seen in the control group. For the three one-handed FCE-OH tests clear cutoff points were defined, for example: for the item trunk movements of the repetitive reaching test lateral flexion movements of the trunk >30 degrees were defined as not WNL. In the rating system we used stills of the videos and added lines in order to make clear what was defined as not within normal limits and within normal limits. Although faces of participants were blurred, we cannot attach the rating system because of privacy reasons. Figure 2 shows an example of one item. We added the rating system without the stills of the videos to the supporting information S4).

Major Revision Request 2.2 – Stratification:

Given the heterogeneity of BPI, stratification by residual function (e.g., ROM, strength categories, lesion levels) is warranted to illustrate item performance across subgroups. This would help clarify whether some items are only reliable in particular clinical subpopulations.

• Response: Analysis per subgroup was not possible because raters only rated 8 out of the 16 videotapes of individuals with BPI each item, in order to prevent prolonged rating sessions. Although we agree that subgroup analyses might have provided interesting additional information, the subgroups would be too small. Furthermore, we think that analysis of the reliability per subgroup is complicated, because there is not a standard measurement tool that classifies individuals with BPI based on residual function, which complicates the determination of the subgroups. Moreover, it would require different systems for which raters need to train and maintain rating skills for different rating systems (see also discussion page 21 line 419- 424).

Major Revision Request 2.3 – Training protocol:

Over half of the raters felt that the video-based training was insufficient. A structured training module (e.g., a set of “gold standard” clips with feedback and short quiz) should be outlined, and the improvement from session 1 to session 2 should be explicitly analyzed as a “learning effect.”

• Response: Thank you for your suggestions, we agree that a structured training module including feedback and practice is needed in further research, this was also suggested in the discussion (page 21, line 408-415).

We suggest that the improvement from session one to session two was caused by a learning effect, but rater specific factors like fatigue, mental health etc. may also have contributed tot the improvement. We did not examine these factors. We added this to the discussion (page 21 line 404-407.

Major Revision Request 3.1 – Kappa limitations:

Low kappa values despite moderate absolute agreement suggest prevalence/bias effects. The authors should include prevalence and bias indices, and/or alternative coefficients such as Gwet’s AC1/AC2 or PABAK.

• Response: The interrater reliability kappa values were determined using Fleiss kappa for multiple raters. Determination of prevalence and bias indices are only available for Cohen’s kappa, not for multiple raters kappa (Byrt et al 1993). Also the PABAK statistic is only described for Cohen’s kappa.

We added the alternative Gwet’s AC1/AC2 coefficient which is suitable for multiple raters to Table 3. The results were similar tot Fleiss kappa, which suggests that there were no prevalence effects. We added this to the Result section, page 15, line 338-339.

We also added the average number of videotapes that were rated as “not within normal limits” per FCE-OH test and overall to Table S5 to illustrate the prevalence rates.

Major Revision Request 3.2 – Advanced modeling:

If feasible, consider multi-level approaches (e.g., Generalizability Theory or mixed-effects logistic regression), which can simultaneously model variance due to rater, video, and session factors.

• Response: Thank you for your suggestions; however we believe that a multi-level approach is not feasible. We initiated the development of the rating system with the ultimate goal of implementing it in clinical practice, provided that reliability is sufficient. In clinical settings, the rating system must be applied by multiple raters to assess multiple individuals with BPI. Therefore, if variance can be attributed to either the videos or the raters, these factors would not be targeted for intervention.

Major Revision Request 3.3 – Item-level behavior:

The weakest performance was for overhead lifting items. These may conflate several compensations (shoulder, trunk, elbow). Splitting them into subcomponents could improve clarity and rater agreement.

• Response: We agree that rating movements of several joints at once may have caused difficulties in rating during these items, we also mentioned this in the discussion (page 20, line 392-397).

In the initials draft of the rating system, joint movements were divided in separate items (see figure S2). However, pilot testing of both the first and second drafts revealed that raters experienced difficulties in evaluating these movements individually which may be caused because movements of several joints were subtle, for example shoulder elevation. Based on our experience developing a rating system for upper limb prosthesis users [19], we learned that subtle deviation movements, such as shoulder elevation, were difficult to asses reliably by observation. In order to rate these more subtle movements, we chose to place greater emphasis on symmetry in the two-handed FCE-OH tests. This adjustment improved the results during pilot testing (page 13, line 293-302).

Revision Request 3.4 – Power and CI width:

Confidence intervals are relatively wide. While the exploratory nature is acknowledged, a brief power calculation or justification of sample size relative to target kappa would add transparency.

• Response: we performed a post hoc power analysis. The sample size was sufficient for the overall intrarater reliability and interrater reliability. The sample size was insufficient for the separate analysis of the inter and intra rater reliability of the items.

We added this to the Results section, page 16 line 340-347:The required sample size for the intra-rater reliability analysis (power 0.8, expected kappa 0.6, precision 0.1) was 16 raters, 10 videos. In our study, 16 raters rated 39 videos, which is a sufficient sample size for the overall intra-rater reliability. However, the sample size for individual items was insufficient because only 7 or 8 videos were rated per item by 16 raters, instead of 10 the videos required to achieve sufficient power.

The required sample size for the interrater analysis (power 0.8, expected kappa 0.6, precision 0.1) was 246 ratings. In our study, each video was rated by 16 raters, resulting in 624 ratings, which is sufficient for the overall interrater reliability analysis. However, for the individual items, only 122 (in case 7 videos were rated) or 128 ratings were obtained, instead of the 246 ratings required to achieve sufficient power.

Several raters reported difficulty due to loss of the sagittal plane in some videos. For future work, multi-angle recordings (front, side, posterior) or camera tracking should be considered.

The instruction that raters could not pause or replay videos mirrors real clinical practice, but should be explicitly justified and included in the user manual.

• Response: We agree that in future studies multi-angle recordings is desired (see also limitation section page 23, line 458-459).

In the instructions the raters received before each rating session, we included that replay or pausing of the videos was not allowed (page 13 line 265-266).

The improvement from session 1 to session 2 suggests a training effect. Quantifying this difference statistically (e.g., confidence intervals for session-to-session change in kappa) would strengthen the claim.

• Response: The width of the confidence intervals for both rating sessions was similar (Table 3), furthermore the confidence intervals of both sessions do overlap. We adjusted text hereabout in the discussion, because there may be a learning effect, but also other unexamined rater specific factors, such as fatigue, mental health, time taken for rating, may have contributed to the improvements between sessions. Page 21 line 405-407.

The interpretation of “almost sufficient” reliability should be replaced with numeric ranges and standard thresholds for clarity.

• Response: We agree that the term “almost sufficient” is not clear. The kappa value of the interrater reliability was moderate for both sessions based on the thresholds we used. We adjusted the abstract and removed the term “almost sufficient”, page 3 line 47-48. In the conclusion we added the threshold for kappa, page 26 line 531-533.

The finding that intra-rater reliability is sufficient while inter-rater reliability is moderate should be highlighted as both a limitation and an opportunity for further rater training.

• Response: Thank you, we added to the discussion session that the interrater reliability may improve with a more extensive training that incorporates structured feedback. Page 21, line 408-409.

6. Data Availability and Ethics

Ethics approval and informed consent are clearly stated.

Data availability statement is appropriate. However, I recommend sharing the raw rating matrix, anonymized survey results, rater instructions, and—if ethically permissible—short example video clips in an open repository with DOI. This will greatly facilitate replication and training.

• Response: Thank you, we will share the raw data and rater instruction after acceptance of the manuscript in an open repository https://doi.org/10.17605/OSF.IO/8JBPY. We will not share the videos because of privacy reasons.

7. Writing, Language, and Presentation

Minor typographical errors should be corrected (e.g., “abled bodied” → “able-bodied,” “prothesis” → “prosthesis”).

Abbreviations should be defined upon first use (e.g., FCE-OH, WNL).

Tables and figures should be polished: abbreviations expanded, statistical measures (κ, CI, % agreement, prevalence) clearly presented.

Supplementary materials should include the final scoring system in a single consolidated PDF.

• Response: Thank you, we adjusted the typing errors. The abbreviations were defined upon the first use. The Tables and Figures were checked, abbreviations were added where necessary. We added the final rating system without illustrative stills of the videos to supporting information S4 (see also our response to comment 2.1).

Reviewer 2

Reviewer #2: 1. Claims & Framing

1.1 Abstract results, discussion and conclusions overstate κ values as “sufficient,” whereas they are best described as moderate (κ≈0.59–0.64).

• Response: We agree the interrater reliability was moderate, therefore we revised “almost sufficient” to moderate, see also our response to comment 3.4 of reviewer 1. Based on the cut of points for kappa we used (poor if ĸ ≤ 0.4, moderate if 0.41< ĸ< 0.59, sufficient if 0.6 ≤ ĸ < 0.79 and good if ĸ ≥0.80) the intrarater reliability was sufficient. The cut off points were based on J. Sim et al. 2005, Page 13 line 281-282.

1.2 The novelty of the tool compared with prior prosthesis-user and observational/postural tools is not sharply positioned in discussion and introduction.

• Response: Previous studies showed that compensatory movements are task specific (page 5 line 100-102) therefore a tool specifically developed for the tasks of the FCE-OH test was needed.

The aim of the developed rating system for individuals with BPI was the same as for upper limb prosthesis users: to assess shoulder and trunk movements during the performance of FCE-OH tests in a standardized and reliable manner. However the rating system for upper limb prosthesis users appeared not to be applicable to individuals with BPI and was therefore adjusted.

Movement patterns of individuals with BPI that were classified as not within normal limits could not be differentiated from normal movement patterns using the previously developed rating system for upper limb prosthesis users.

In the introduction we added that to the best of our knowledge an observational rating system rating movement patterns applicable for the use in individuals with BPI does not exist, in order to position the novelty of the rating system better (page 6 line 109-110).

In the discussion we added that a new rating system was developed and clarified that the rating system designed for prosthesis wearers was not applicable to individuals with BPI. Page 19 line 374 and page 19 line 380-382.

2. Scale Development (Methods)

2.1 The reliance on one student, one professional, and internal group consensus limits the robustness of the item selection; broader expert involvement (e.g., multiple clinicians or a Delphi panel) would improve validity.

• Response: The internal consensus group consisted of 3 rehabilitation physicians and one FCE-expert (this was added to the method section page 11, 228-230). Additionally, raters added suggestions for improvement of items or extra items in the feedback during the construction phase. These suggestions were used to improve the items of rating system. We agree that a Delphi study may result in a rating system with enhanced validity. We added this to the discussion, page 23 line 470-472.

2.2 The choice of a >2 SD threshold is statistical but not clearly justified clinically. Why is this cutoff appropriate for defining abnormal compensation?

• Response: We agree that the threshold was as statistical choice, see also our response to comment 2.1 of reviewer 1. Determining a clinically relevant threshold is challenging, as to our knowledge only three previous studies have examined movement patterns in individuals with BPI [2,4]. These studies demonstrated that movement patterns were task specific. None of these studies analyzed movement patterns during FCE-OH tasks or comparable activities.

The chosen threshold may have resulted in that only larger movements were classified as not within normal limits. Clinically we think these movements are the most important to intervene, because these movements cause larger internal moments around joints and require therefore more muscle force. These movements may therefore cause muscle fatigue and thereby increase the risk of developing musculoskeletal complaints [5]. We added this to the limitation section of the discussion, page 22 and 23 line 448-455.

2.3 The binary WNL/nWNL scoring system is simple and p

---

## [Editor Report · Decision Letter 1]

5 Mar 2026

Development and reliability testing of a qualitative observational rating system for individuals with brachial plexus injury performing Functional Capacity Evaluation tests.

PONE-D-25-38741R1

Dear Dr. Tallie van der Laan,

We’re pleased to inform you that your manuscript has been judged scientifically suitable for publication and will be formally accepted for publication once it meets all outstanding technical requirements.

Kind regards,

Priti Chaudhary, M.S.

Academic Editor

PLOS One
---

## [Editor Report · Acceptance letter]

PONE-D-25-38741R1

PLOS One

Dear Dr. van der Laan,

I'm pleased to inform you that your manuscript has been deemed suitable for publication in PLOS One. Congratulations! Your manuscript is now being handed over to our production team.

Kind regards,

on behalf of

Dr. Priti Chaudhary

Academic Editor

PLOS One